# Potential of Modulating Aldosterone Signaling and Mineralocorticoid Receptor with microRNAs to Attenuate Diabetic Kidney Disease

**DOI:** 10.3390/ijms25020869

**Published:** 2024-01-10

**Authors:** Shinji Hagiwara, Tomohito Gohda, Phillip Kantharidis, Jun Okabe, Maki Murakoshi, Yusuke Suzuki

**Affiliations:** 1Department of Nephrology, Juntendo University Faculty of Medicine, Tokyo 1138421, Japan; maki-m@juntendo.ac.jp (M.M.); yusuke@juntendo.ac.jp (Y.S.); 2Hagiwara Clinic, Tokyo 2030001, Japan; 3Department of Diabetes, Monash University, Melbourne, VIC 3004, Australia; phillip.kantharidis@monash.edu (P.K.); jun.okabe@baker.edu.au (J.O.); 4Epigenetics in Human Health and Disease Program, Baker Heart & Diabetes Institute, Melbourne, VIC 3004, Australia

**Keywords:** Diabetic Kidney Disease, microRNA, aldosterone, mineralocorticoid receptor

## Abstract

Diabetic Kidney Disease (DKD) is a significant complication of diabetes and primary cause of end-stage renal disease globally. The exact mechanisms underlying DKD remain poorly understood, but multiple factors, including the renin–angiotensin–aldosterone system (RAAS), play a key role in its progression. Aldosterone, a mineralocorticoid steroid hormone, is one of the key components of RAAS and a potential mediator of renal damage and inflammation in DKD. miRNAs, small noncoding RNA molecules, have attracted interest due to their regulatory roles in numerous biological processes. These processes include aldosterone signaling and mineralocorticoid receptor (MR) expression. Numerous miRNAs have been recognized as crucial regulators of aldosterone signaling and MR expression. These miRNAs affect different aspects of the RAAS pathway and subsequent molecular processes, which impact sodium balance, ion transport, and fibrosis regulation. This review investigates the regulatory roles of particular miRNAs in modulating aldosterone signaling and MR activation, focusing on their impact on kidney injury, inflammation, and fibrosis. Understanding the complex interaction between miRNAs and the RAAS could lead to a new strategy to target aldosterone signaling and MR activation using miRNAs. This highlights the potential of miRNA-based interventions for DKD, with the aim of enhancing kidney outcomes in individuals with diabetes.

## 1. Introduction

The International Diabetes Federation reports that in 2021, there were 537 million individuals globally with diabetes. This figure is expected to rise to 783 million by 2045. Diabetic Kidney Disease (DKD) is highlighted as a significant complication linked to diabetes and is one of the primary causes of end-stage renal disease (ESRD). Approximately 30–40% of individuals diagnosed with diabetes will ultimately develop DKD. Although the exact mechanism leading to the development of DKD is still unclear, it is known that several factors, besides hyperglycemia, contribute to its onset. These factors include genetic, environmental, and hemodynamic elements such as hypertension, aging, arteriosclerosis, dyslipidemia, and proteinuria. DKD typically initiates with microalbuminuria (30–300 mg of albumin per day) and advances to macroalbuminuria (over 300 mg) in later stages. Albuminuria not only indicates DKD but also serves as an independent risk factor for cardiovascular disease. Herein, it is crucial to emphasize that individuals with DKD have a higher risk of mortality than progressing to ESRD. DKD progression traditionally involves a sequential increase in albuminuria, followed by a decrease in glomerular filtration rate (GFR), ultimately leading to ESRD. However, a growing body of evidence suggests a profound transformation in the natural course of DKD over the past few decades owing to the evolving landscape of diabetes, advancements in therapeutic interventions, and aging of the general population [1]. The intricate interplay of metabolic, epigenetic, and transcriptional factors contributes to the development of DKD. Hyperglycemia triggers increased production of advanced glycation end products and reactive oxygen species (ROS), thus resulting in oxidative stress. Elevated glucose levels induce ROS production, thereby impacting mitochondrial function and causing DNA damage. The ROS activation of Janus kinase/signal transducer and activator of transcription protein (JAK-STAT) pathways results in inflammation. Furthermore, high glucose levels stimulate the hexosamine pathway, thus influencing vascular damage. Hyperglycemia initiates podocyte loss, glomerular fibrosis, and abnormal wound healing processes. Fibrosis is associated with transforming growth factor-β1 (TGF-β1) signaling in mesangial cells and activation of Wnt/β-catenin signaling [2]. Additionally, the role of low-grade chronic inflammation as a key contributor to pathological signaling has been increasingly recognized. A wide range of vasoactive hormones, growth factors, and inflammatory cytokines mediate progressive renal disease. 

Among these, the renin–angiotensin–aldosterone system (RAAS) plays a pivotal role with clear benefits through the use of inhibitors targeting this pathway in patients with albuminuria. The RAAS is a well-known hormonal cascade that leads to the release of the mineralocorticoid steroid hormone, aldosterone. This release is activated by a decrease in plasma sodium (Na^+^), an increase in circulating potassium (K^+^), or a reduction in effective circulating volume [3]. The RAAS serves as a vital homeostatic feedback mechanism. It coordinates the functions of renal and vascular tissues, regulating blood volume and aiding in the maintenance of blood pressure [3,4]. Aldosterone is generated in the zona glomerulosa cells within the adrenal glands, as a response to renin [5,6,7,8]. Besides its crucial role in maintaining body fluid balance, aldosterone is also suggested as a potential contributing factor in the progression of renal disease. It is associated with kidney inflammation and fibrosis, which can result in conditions such as renal hypertrophy, tubulointerstitial fibrosis, and glomerulosclerosis [9,10,11,12,13]. Currently, ACE inhibitors (ACEI) and Ang II receptor blockers (ARB) are used to slow the progression of DKD. Nonetheless, certain clinical trials have demonstrated that due to the phenomenon known as “aldosterone escape”, characterized by an increase in plasma aldosterone levels after an initial decrease with ACEI/ARB treatment, the efficacy of ACEI or ARB in reducing urinary albumin may not meet expectations in some patients [14,15]. Increasing evidence suggests that hyperaldosteronism contributes to renal damage [16]. In the general population, an inverse association has been found between plasma aldosterone concentration and the GFR [17]. Additionally, an enhanced decline in the GFR in patients with DKD has been associated with aldosterone escape during long-term blockade of the RAAS system [18]. 

Aldosterone attaches to the mineralocorticoid receptor (MR), which then triggers the transcription of proteins. These proteins collaborate to create a coordinated genomic response within the cell [19]. The activation of the MR contributes to the propagation of kidney injury, inflammation, and fibrosis, thereby playing a role in the progression of chronic kidney disease (CKD) [20]. The nonsteroidal mineralocorticoid receptor antagonist (MRA), finerenone, has demonstrated effectiveness in slowing the progression of kidney disease in patients with type 2 diabetes mellitus [21,22]. Consequently, the U.S. Food and Drug Administration has recently granted approval for the use of finerenone for this indication. Moreover, the recently published annual guidelines from the American Diabetes Association, *Standards of Medical Care in Diabetes*, recommend the use of finerenone to slow the progression of CKD and reduce cardiovascular events [23].

MicroRNAs (miRNAs) are small RNA molecules that are naturally present in cells and do not code for proteins. Typically, miRNAs consist of 18–22 nucleotides and originate from DNA. The initial transcripts of miRNAs, known as primary miRNA transcripts (pri-miRNAs), undergo processing in the cell nucleus by a complex involving Drosha (an RNase III enzyme) and DGCR8 (DiGeorge syndrome critical region gene 8). This processing transforms pri-miRNAs into pre-miRNAs, which adopt a characteristic hairpin-loop structure [24]. To move from the nucleus to the cytoplasm, pre-miRNAs utilize exportin 5. Once in the cytoplasm, they undergo further processing by another RNase III enzyme called Dicer. The role of Dicer is to cleave the terminal loop of the pre-miRNA, which results in the creation of a mature miRNA duplex, typically approximately 22 base pairs long. One strand from this miRNA duplex is selected and incorporated into the RNA-induced silencing complex (RISC), whereas the other strand is degraded. The RISC–miRNA complex recognizes the 3′-untranslated region (UTR) of the target mRNA through partially complementary nucleotide sequences, ultimately leading to the degradation of the target mRNA or translational repression and thereby controlling gene expression [25,26,27]. A single miRNA has the ability to regulate the expression of numerous genes within signaling pathways, which influence multiple biological pathways and cellular functions and potentially contribute to disease development [28,29].

The interaction between metabolic and hemodynamic pathways, including hypertension, the RAAS, and vasoactive hormones, significantly contributes to the development and progression of DKD [30]. Additionally, we have previously examined the role of miRNA in relation to the metabolic and hemodynamic pathways that contribute to the progression of DKD and also explored the potential of targeting renal fibrosis using miRNA [24,31]. Several miRNAs have been reported to play a role in aldosterone signaling and MR expression. As MR is a nuclear transcription factor, it modulates the transcription of miRNAs relevant to DKD. In this review, we outline the functions of miRNAs in the propagation of kidney injury, inflammation, and fibrosis through aldosterone signaling and MR activation. In Table 1, we present a list of miRNAs along with their relevant targets associated with renal disease. The functions of these miRNAs are summarized in Figure 1 and outlined below. While not all miRNAs in Table 1 are specific to DKD, the compilation focuses on those that are associated with aldosterone signaling and MR activation to highlight their potential relevance. Recent clinical trials demonstrating the efficacy of an MR antagonist in slowing down the progression of renal dysfunction in type 2 diabetes support the exploration of miRNAs associated with aldosterone signaling and MR activation as potential novel therapeutic targets for DKD. 

## 2. Function of miRNA

### 2.1. MiR-192

High blood pressure is associated with high expression of *WNK1*, a gene that plays an important role in regulating electrolyte transport in the kidney. The 3′ UTR of *WNK1* contains target sequences for two miRNAs, miR-192 and miR-215, and the functions of both have been validated. While both microRNAs were expressed in the distal tubules, miR-192 was determined to be the only regulator of *WNK1,* specifically its long form (L-WNK1). The regulation of miR-192 is influenced by sodium (Na^+^) and potassium (K^+^) levels, as well as aldosterone levels. Notably, the administration of aldosterone was found to decrease miR-192 levels in the kidney [32]. In vitro experiments conducted on MDCK cells confirmed the role of miR-192 in regulating *WNK1*. Furthermore, computational analysis revealed that miR-192 has the ability to directly regulate ion transporters and channels, including *CLC5* and the α1-subunit of Na^+^–K^+^ ATPase [32]. These components are crucial for maintaining the transmembrane ion gradient in the renal distal tubules [33].

### 2.2. MiR-802

In studies involving rodent models, it was found that a high dietary intake of K^+^ stimulated the transcription and expression of miR-802 in the collecting ducts of the kidney. Consequently, this increase in miR-802 resulted in a decrease in caveolin-1 (*CAV1*) levels through a specific miR-802 seed sequence in the 3′ UTR of the *CAV1* gene. As a consequence, the internalization of renal outer medullary potassium (ROMK) channels was reduced [34]. The regulation of *CAV1* by miR-802 was confirmed through in vitro experiments, which demonstrated that a decrease in endogenous miR-802 led to an increase in *CAV1* levels. Essentially, miR-802 enhances the impact of a high-potassium diet on ROMK channel activity by suppressing *CAV1* expression. This suppression results in an increased surface expression of ROMK channels in the distal nephron [34].

### 2.3. MiR-194

The same research group conducted a study demonstrating that increased intake of dietary K^+^ also influences the regulation of ROMK channels through miR-194. Higher dietary K^+^ levels increase miR-194 expression, whereas low K^+^ levels decrease miR-194 levels. It was observed that the correlation between aldosterone release and miR-194 responded to dietary K^+^ levels. In addition, the researchers found that miR-194 reduces the expression of intersectin-1 (encoded by the *ITSN1*), which is a scaffold and regulatory protein. The reduction in intersectin-1 expression increases the presence of ROMK channels on the cell surface. This phenomenon is induced by delayed removal of ROMK channels from the plasma membrane, ultimately enhancing K^+^ transport [35]. Therefore, changes in dietary K^+^ intake provide two distinct mechanisms for controlling K^+^ transport by modulating different miRNAs, specifically miR-802 and miR-194, which ultimately lead to similar outcomes.

### 2.4. MiR-20a

11β-Hydroxysteroid dehydrogenase type 2 (11β-HSD2) is expressed in aldosterone-sensitive tissues and allows these tissues to respond to aldosterone through MR. Regulation of 11β-HSD2 expression is complex and involves CpG methylation, a number of transcription factors, and multiple microRNA binding sites. A reduction in 11β-HSD2 activity leads to MR activation, which results in sodium retention and salt-sensitive hypertension. To study the involvement of miRNA, two rat models with different 11β-HSD2 activity levels were compared: Sprague–Dawley rats with low activity and Wistar rats with high activity. Notably, variations in the expression of rno-miR-20a-5p were observed between these groups. The observed variation is significant because the 3′ UTR of *HSD11B2* contains a binding site for rno-miR-20a-5p. The introduction of a miR-20a mimic confirmed the presence and function of this binding site by reducing the activity of 11β-HSD2 in HT29 and SW620 cells [36]. However, miR-20a also regulates the E2F transcription factor, the levels of which rapidly change due to renal ischemia [37,38]. Therefore, it remains unclear whether the reduction in 11β-HSD2 activity is a direct result of miR-20a interacting with the 3′ UTR of *HSD11B2* mRNA alone or an indirect consequence through the regulation of the *E2F* transcription factor.

### 2.5. MiR-23~24~27

The epithelial sodium channel (ENaC) plays a pivotal role in balancing Na^+^ levels and water in the kidney. Aldosterone primarily controls ENaC regulation in collecting duct epithelial cells. A recent study found that the levels of a group of miRNAs, mmu-miR-23–24–27, increase in response to aldosterone stimulation in the cortical collecting duct (CCD) of kidney nephrons [39]. Further investigations showed that miR-27a/b, members of this miRNA cluster, bind to the 3′ UTR of *ITSN2* encoding intersectin-2, a protein in the distal kidney nephron that regulates membrane trafficking [39]. Interestingly, the introduction of miR-27 and other members of this family into mouse CCD cells, even without aldosterone stimulation, also increases ENaC activity [39]. This increase in activity likely occurs due to the reduction of *ITSN2* expression, thus preventing ROMK internalization from the cell surface, as previously described.

Aldosterone induces miR-24 in MR-sensitive cells, including the epithelial cells of the distal nephron [39], and negatively regulates aldosterone production by targeting aldosterone synthase [40]. An analysis of healthy human adrenal tissue and aldosterone-producing adenoma showed distinct miRNA expression patterns, especially with significant differences in the expression of miR-24. Potential binding sites for miR-24 were found in the 3′ UTR of *CYP11B1* (11β-hydroxylase) and *CYP11B2* (aldosterone synthase) mRNAs. In vitro experiments confirmed that miR-24 can regulate *CYP11B1* and *CYP11B2* expression, thereby affecting cortisol and aldosterone [40]. Similarly, aldosterone-induced miR-27a targets the angiotensin-converting enzyme [41]. Therefore, the aldosterone-stimulated miR-23~24~27 family can theoretically decrease the expression of two crucial enzymes for aldosterone production, thereby creating an extended feedback loop.

### 2.6. MiR-34c

Park et al. studied fibrotic signaling driven by the MR in the kidney and detected fibrosis-related proteins in mouse cortical duct cells (mCCD) upon aldosterone treatment [42]. They studied alterations in miRNA expression after aldosterone treatment and observed a number of changes, including in 15 miRNAs closely associated with the regulation of the Wnt signaling pathway. Further investigation revealed that miR-34c-5p specifically targeted Ca^2+^/calmodulin-dependent protein kinase type II beta-chain (CaMKIIβ). In aldosterone-treated cells, the reduction in miR-34c-5p levels increased CaMKIIβ mRNA and protein and stimulated fibronectin (FN) and alpha-smooth muscle actin. Transfection of mCCD with either *CAMK2B* siRNA or the miR-34c-5p mimic significantly reduced the FN induction by aldosterone, which was accompanied by decreased CaMKIIβ protein levels. These findings confirm the involvement of miR-34c-5p and CaMKIIβ in aldosterone-induced fibrosis. A luciferase reporter assay validated the direct effect of miR-34c-5p on CaMKIIβ mRNA and protein expression. In conclusion, their study suggests that the downregulation of miR-34c-5p, induced by aldosterone in the Wnt signaling pathway, contributes to the increased expression of CaMKIIβ. This plays a significant role in the development of fibrosis induced by aldosterone [42].

### 2.7. MiR-196b

In db/db mice with diabetes, it was demonstrated that aldosterone increased proteinuria and the accumulation of tubulointerstitial extracellular matrix (ECM) [43]. However, the administration of MRA eplerenone alleviated the adverse effects of aldosterone. In coculture experiments, aldosterone-treated proximal tubular epithelial cells (PTECs) released extracellular vesicles (EVs). When these EVs were taken up by renal fibroblasts, an enhancement in ECM production was observed. Furthermore, injecting EVs from aldosterone-treated db/db mice into C57BL/6 mice resulted in increased ECM accumulation in the kidney. Subsequent miRNA analysis revealed that miR-196b-5p was the most elevated miRNA in EVs derived from PTECs after aldosterone stimulation. Further experiments revealed that overexpression of miR-196b-5p in fibroblasts increased ECM production, decreased suppressor of cytokine signaling 2 (SOCS2) expression, and resulted in elevated phosphorylation of the signal transducer and activator of transcription 3 (STAT3). In patients with DKD, the plasma levels of miR-196b-5p were found to be elevated, correlating with albuminuria levels. Elevated levels of miR-196b-5p were predominantly detected in PTECs in kidney samples from patients with DKD. This study identified a novel mechanism of aldosterone-induced kidney damage in diabetes through the miR-196b-5p-EV pathway. This mechanism involves communication between PTECs and fibroblasts, which is potentially mediated by STAT3/SOCS2 signaling [43].

### 2.8. MiR-766

A recent study demonstrated that miR-766 directly targets the MR gene *NR3C2*, thus promoting proliferation and metastasis in hepatocellular carcinoma [44]. The level of miR-766 correlates with the prognosis of liver cancer in such patients. In a different study, miR-766-3p served as a crucial regulator of inflammatory response in patients with rheumatoid arthritis (RA). Further analysis of plasma samples from patients with RA treated with abatacept revealed eight differentially expressed miRNAs, including miR-766-3p. MiR-766-3p indirectly reduced the activation of NF-κB; the same mechanism was involved in the reduction of MR expression. The anti-inflammatory effect of miR-766-3p was not limited to a specific cell type but suppressed inflammatory gene expression in various cell types, including human RA fibroblast-like synoviocytes (MH7A) and primary normal human mesangial cells (NHMCs) [45]. These findings indicate that miR-766-3p can offer a new therapeutic strategy for inflammation reduction. This approach may have potential applications not only for RA but also for kidney diseases.

### 2.9. MiR-466 Family

Another study investigated the increased expression of miRNA in mouse kidney after prolonged exposure to aldosterone for more than 3 days [46]. MiR-466a-e was found to be responsive to aldosterone, showing elevated levels both in vitro and in vivo in mCCD cells after aldosterone treatment. This miRNA family is known to target and regulate MR and *SGK1* expression, consistent with the presence of target sites in the 3′ UTRs of these genes. Inhibition of these miRNAs significantly enhanced the expression of MR and *SGK1*, thus enhancing aldosterone sensitivity in mCCD. These findings indicated a localized feedback loop in MR signaling, thus demonstrating that aldosterone-induced miRNAs effectively target both the receptor and its downstream effector, *SGK1*. This mechanism helps prevent excessive signaling from prolonged aldosterone exposure [46].

### 2.10. MiR-324-5p and MiR-30

Variations in extracellular tonicity across different nephron segments influence MR expression through mechanisms that are still not well understood [47]. Several RNA-binding proteins play a role in the posttranscriptional regulation of MR in response to extracellular tonicity. MiR-324-5p and miR-30c-2-3p, which are upregulated in renal cells under hypertonic conditions and in the kidneys of mice treated with furosemide, decrease MR expression [48]. These miRNAs directly affect the stability of MR transcripts by targeting *NR3C2* and collaborate with tetradecanoyl phorbol acetate inducible sequence 11b (TIS11B) to degrade MR mRNA. In addition, these miRNAs inhibit *ELAVL1* (HuR: Human antigen R) transcripts, consequently enhancing MR expression and signaling. The overexpression of miR-324-5p and miR-30c-2-3p modifies MR expression and signaling in KC3AC1 cells and attenuates aldosterone-regulated gene expression. Furthermore, their expression increases in hypertonic mouse kidneys after furosemide treatment [48]. These findings shed light on the role of MR signaling in nephropathies, including diabetic nephropathy [49] and partial aldosterone resistance in newborns [50].

### 2.11. Tug1-miR-29b

MiR-29b-3p has the potential to inhibit epithelial-to-mesenchymal transition (EMT) and fibrosis by regulating gene expression associated with the ECM in the kidney [51,52]. Cesana et al. [53] demonstrated a novel mechanism in which long noncoding RNAs (lncRNAs) control gene expression. LncRNAs serve as competing endogenous RNAs (ceRNAs), which bind to and sequester miRNAs, thereby inhibiting their interaction with target messenger RNAs (mRNAs). Bioinformatics analysis revealed that lncRNA Tug1 functions as a ceRNA and directly interacts with and sequesters miR-29b-3p. Furthermore, the study demonstrated that MR directly interacts with lncRNA Tug1, thus providing insight into the mechanisms of renal fibrosis involving MR and lncRNA signaling [54]. Moreover, the study demonstrated increased Tug1 in human renal biopsy samples correlating with enhanced fibrosis. Interestingly, the expression of Tug1 increases in renal cells exposed to angiotensin II and hypertensive nephropathy. This potentially establishes a connection between MR and renal fibrosis through the modulation of miR-29b-3p via ceRNAs [54].

### 2.12. MiR-21

MiR-21, which is one of the most extensively studied miRNAs, plays a significant role in various renal diseases, such as CKD, diabetic nephropathy, renal cell carcinoma, and renal fibrosis [55]. It is consistently upregulated in all these diseases [56,57,58,59,60]. Furthermore, miR-21 exacerbates renal fibrosis by enhancing profibrotic TGF-β signaling. It targets negative regulators of fibrosis, mothers against decapentaplegic homolog 7 (SMAD7), and phosphatase and tensin homolog (PTEN), thus leading to increased expression of ECM genes and other fibrotic factors [61]. In a study conducted on hypertensive mice, a treatment using deoxycorticosterone acetate (DOCA), a precursor of aldosterone, resulted in an increased incidence of kidney injury. MiRNA profiling revealed elevated levels of miR-21 in the kidneys and urine from diseased kidneys compared with the controls. Interestingly, urinary miR-21 was detected earlier than albumin and showed a correlation with kidney damage markers. Assessment of urinary miR-21 levels may thus offer a noninvasive tool for detecting hypertensive kidney injury [62]. The urinary miRNA-21 biomarker holds potential for improved disease monitoring and diagnosis. Moreover, it can potentially serve as a therapeutic target in clinical trials for hypertensive renal injury and fibrosis.

### 2.13. MiRNA-Based Therapeutic Strategies

Increased understanding of microRNAs in biology and their dysregulation in various diseases has prompted scientists to explore their potential for therapeutic applications. Two strategies have been employed. One strategy involves miRNA restoration therapy, where a downregulated or nonfunctional miRNA is supplied by a synthetic oligonucleotide, whereas the other strategy involves miRNA inhibition therapy, where miRNA overexpression is inhibited by antagonists [63]. 

MiRNA mimics used for therapeutic purposes are engineered to replicate the functions of endogenous miRNA. These synthetic double-stranded oligonucleotides undergo cellular processing to imitate the natural function of miRNA, thus offering enhanced stability and chemical modifications for efficient delivery and entry into target cells. Contrarily, inhibition of endogenous miRNA can be achieved by introducing anti-miRNA/ antagomiR oligonucleotides (AMOs). AMOs target pri-miRNA, pre-miRNA, or mature miRNA to either sequester or eliminate endogenous miRNAs [64]. MiRNAs are generally stable; however, certain cellular environments can lead to rapid decay of individual miRNAs [65]. Thus, to enhance RNA stability in vivo, various modifications have been implemented, such as (1) replacement of a phosphodiester backbone with a phosphorothioate backbone, (2) modification of the ribose 2′-OH group, (3) incorporation of locked nucleic acid (LNA) modifications, and (4) application of peptide nucleic acid modification [28,66].

There are various methods for delivering miRNAs, such as conjugation [67], virus-associated delivery [68], and nanoparticles [69]. Virus-associated miRNA delivery has shown experimental efficiency in cancers; however, safety concerns associated with viruses have limited its clinical application. Nonviral delivery systems appear more promising. The conjugation system, where lipids or cell receptor-targeting ligands are directly attached to miRNA, offers a potentially safer approach for miRNA delivery. However, inadequate dispersion can lead to miRNA accumulation in the liver, necessitating the requirement of high doses for sufficient delivery and thus limiting their applications. An example is RG-101, an anti-miRNA-122 covalently conjugated with N-acetylgalactosamine and developed for hepatocyte delivery, which is currently in clinical trials [70]. Nanoparticles have several advantages for miRNA delivery, including a cationic component that complexes with anionic miRNAs to protect them from degradation and facilitate their cellular uptake [71].

While miRNA-based drugs are currently unavailable, numerous miRNA-based therapies for various human diseases are in clinical trials. These include Miravirsen (an LNA and phosphorothioate-modified antagomiR targeting miRNA-122), RG-101 (an N-acetyl-D-galactosamine-conjugated antagomiR targeting miR-122), RG-125 (an antagomiR targeting miR-103/107), RGLS5040 (an anti-miR-27), RG-012 (a phosphorothioate, 2′-O-methoxyethoxy-modified antagomiR targeting miR-21), MRG-201 (an LNA miRNA-29b mimic), MRX34 (an miRNA-34a mimic), MRG-106 (an LNA antagomiR targeting miR-155), MRG-107 (an antagomiR targeting miR-155), MRG-110 (an LNA antagomiR targeting miR-92), MesomiR (an miR-16 mimic), and ABX464 (a small molecular compound triggering miR-124 expression) [72]. Some of these miRNA-based therapies, such as Miravirsen and RG-101 targeting miR-122, have shown efficacy against hepatitis C [70,73]. However, challenges such as severe side effects have led to the suspension of the development of certain therapies, including RG-101, RG-125, and RGLS5040 [73,74,75,76]. In addition, miRNA mimics, such as MRG-201 for miR-29 and MRX34 for miR-34, aim to restore levels of the respective miRNAs in diseases like fibrosis and cancer [77,78]. Other compounds, such as MRG-106, MRG-107, MRG-110, Mesomir, and ABX464, are being explored for their therapeutic potential in lymphoma, leukemia, amyotrophic lateral sclerosis, ischemic conditions, malignant pleural mesothelioma, and inflammatory bowel disease, with some already in clinical trials [79,80,81,82,83,84,85,86].

As our understanding in this field advances, the goal is to seamlessly integrate the precise adjustment of specific miRNAs into well-established clinical approaches, with the aim of establishing a proactive and efficient strategy for nephroprotection in the near future.

## 3. Discussion

There has been a growing interest in targeting the aldosterone–MR axis as a potential approach for DKD. Modulating aldosterone signaling and the MR using miRNAs to attenuate DKD progression is a highly promising, novel, and innovative approach. It not only has the potential to revolutionize kidney disease management but also challenges our current understanding of the underlying mechanisms at play in kidney disease. The intricate research and findings presented in this review highlight these mechanisms and support the view that miRNAs play a pivotal role in orchestrating the factors driving DKD progression and mediating the profound influence of aldosterone on renal function.

MicroRNA-based therapy remains to have many hurdles that it needs to overcome in terms of targeted delivery; however, it also holds the remarkable potential to be highly precise in targeting specific molecular pathways that are central to DKD. By selectively modulating aldosterone signaling and MR expression, treatments may be tailored to individual patients, taking their unique genetic and miRNA profiles into consideration. This approach offers a departure from the one-size-fits-all paradigm toward therapies and promises enhanced efficacy with fewer side effects.

DKD is typically characterized by extensive renal fibrosis and inflammation, both of which are major factors contributing to the development and progression of kidney damage. Many of the aforementioned miRNAs are intricately involved in the regulation of fibrotic and inflammatory processes in the kidney. Modulation of these miRNAs holds promise for better outcomes for patients suffering from this insidious condition by reducing fibrosis and inflammation and potentially slowing down or even reversing DKD progression. In comparison, traditional treatments such as ACEI and ARBs are often associated with side effects, such as hyperkalemia or acute kidney injury, and patients may not optimally respond due to aldosterone escape. MicroRNA-based therapy therefore presents an alternative to current therapies with a different mechanism of action, thus suggesting the dawn of a new era of therapeutics and the possibility of an alternative path to recovery.

Some of the many studied miRNAs, such as miR-21, have shown immense promise as biomarkers for kidney damage. This potential for early detection and timely diagnosis of DKD is often a critical turning point for patients. It not only enables a swift and precise intervention but also has the potential to guide more personalized patient stratification. Different miRNAs may emerge as potent tools, each with unique efficacy, thus offering the prospect of finely tailored treatments for distinct subgroups of patients with DKD.

Despite the many promising developments in this area of research, it is important to acknowledge that miRNA-based therapies are still in the experimental stage. Comprehensive preclinical and clinical studies are warranted to unveil the full spectrum of their safety and efficacy. In addition, addressing the challenge of developing delivery mechanisms and ensuring that miRNAs reach specific kidney cells remains an important issue.

In summary, the prospect of using miRNAs to modulate aldosterone signaling and MR to mitigate DKD is indeed an exciting avenue of research. It offers the potential for more personalized, targeted, and highly effective treatments. However, the path forward still requires further research, comprehensive clinical trials, and collective efforts of the scientific community to fully unlock this therapeutic potential.

## 4. Conclusions

DKD is a major microvascular complication of diabetes and the primary cause of ESRD. The RAAS, especially aldosterone, is important in managing electrolyte and fluid balance by acting on kidneys. Dysregulation of aldosterone leads to kidney fibrosis and inflammation in DKD through a number of complex mechanisms. The application of MRAs has demonstrated potential as a new treatment for DKD. In parallel, recent studies have concentrated on miRNA profiling to more accurately define the intricate and interconnected pathways that lead to the development and progression of DKD, with the goal of developing new miRNA-based therapies. Investigating the relation among miRNAs, the aldosterone, and DKD will offer insights into the molecular mechanisms involved in this complex diabetes complication, thus potentially paving the way for new therapeutic strategies.

## 5. Future Directions

The promising role of miRNAs in regulating aldosterone signaling and MR expression in the context of DKD warrants further exploration. Validation of these early findings in preclinical models is an important next step with a view to ultimately testing this approach in clinical trials. Such a validation is crucial to demonstrate the safety and efficacy of miRNA-based interventions in patients with DKD.

There has been a considerable interest in the identification of improved miRNA biomarkers for the early detection of kidney damage in DKD. Specific miRNAs, such as miR-21, exhibit great potential as biomarkers for DKD. To improve disease management, further research is warranted, and reliable and early diagnostic tools need to be developed based on these miRNAs. Such easily accessible biomarkers in urine or blood could revolutionize the diagnosis of DKD and enable timely interventions.

For miRNAs to be used therapeutically, it is crucial to develop effective delivery mechanisms to achieve successful translation to clinical applications. Several novel delivery methods are being developed for different diseases. The need to develop delivery methods targeting specific renal cell populations remains an important challenge for achieving the desired therapeutic outcomes.

Exploring the potential of combining miRNA-based therapies with existing treatments, such as ACEi and ARBs, opens exciting possibilities. Further research is warranted to determine whether such combinations can exert synergistic effects, thus potentially addressing the issue of aldosterone escape. With a view to navigating a path toward clinical use, engagement with regulatory authorities and health agencies is essential, and understanding of the requirements for approval and the establishment of safety and efficacy standards for miRNA-based therapies are vital steps in this journey.

Promoting collaboration among researchers, clinicians, and pharmaceutical companies is crucial for the development and implementation of miRNA-based therapies, and such a multidisciplinary approach is important for advancing in this field. This review lays a strong case in favor of the development of miRNA-based interventions in DKD. The future directions highlighted here are aimed at transforming these findings into tangible clinical applications, thus providing new hope to individuals with diabetes and those at risk of DKD. This innovative approach holds promise for more precise, effective, and personalized treatments for this complex and challenging condition.

## Figures and Tables

**Figure 1 ijms-25-00869-f001:**
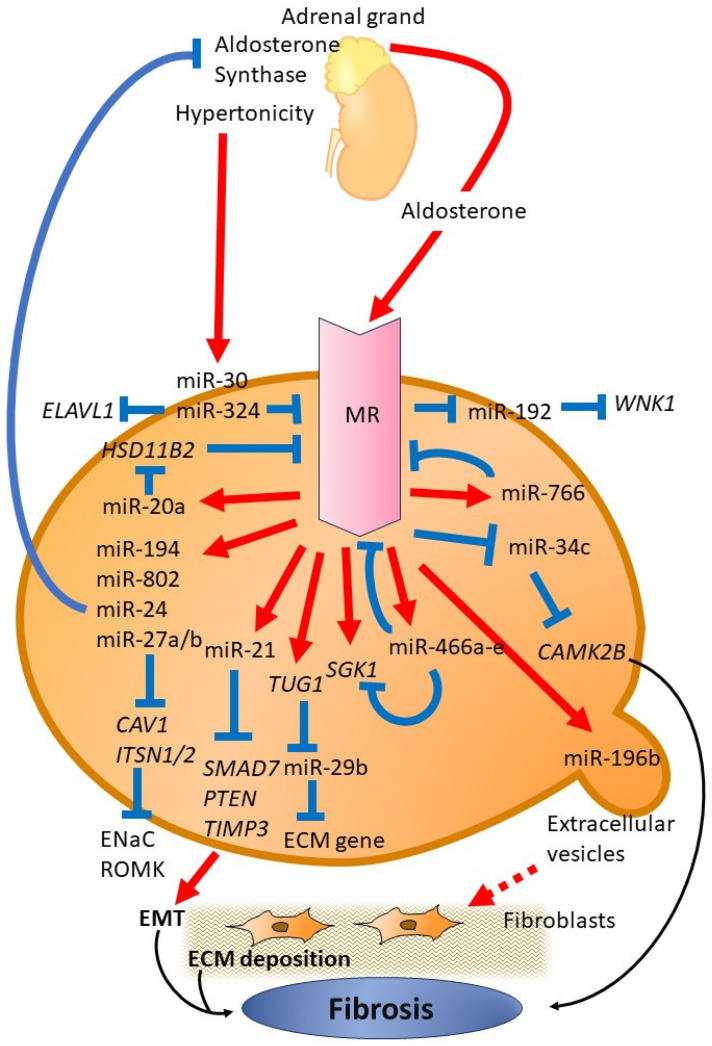
A schematic diagram depicting the miRNA signaling pathways activated by aldosterone through the MR. These miRNAs impact different aspects of the aldosterone and MR pathway and subsequent molecular processes, which affect sodium balance, ion transport, and the regulation of fibrosis. Red lines represent enhanced expression, whereas blue arrows denote repression. Several miRNAs induced by aldosterone might provide negative feedback to regulate components of the RAAS signaling pathway. MR, mineralocorticoid receptor; ROMK, renal outer medullary potassium; ENaC, epithelial sodium channel; ECM, extracellular matrix; EMT, epithelial–mesenchymal transition.

**Table 1 ijms-25-00869-t001:** This table summarizes the role of miRNA in aldosterone signaling and its association with the mineralocorticoid receptor and renal disease.

miRNA	Targeted Gene	Status	Affected Cells	Mechanism	Reference
miR-192	*WNK1*	Reduced	Distal convoluted tubules, connecting tubules, cortical collecting duct cells	Regulating ion transporters and channels	[32,33]
miR-802	*CAV1*	Overexpressed	Cortical collecting duct cells	Increased due to high K^+^ intake Enhancing the ROMK1 channel	[34]
miR-194	*ITSN1*	Overexpressed	Cortical collecting duct cells	Increased due to high K+ intake Extending the presence of ROMK channels on the cell surface delays their removal from the plasma membrane, thereby enhancing K+ transport	[35]
miR-20a	*HSD11B2* *E2F1, E2F2, E2F3*	Overexpressed	Cortical collecting duct cells	Reduced activity of 11β-HSD2 results in the overactivation of the mineralocorticoid receptor	[36,37,38]
miR-27a/b	*ITSN2*	Overexpressed	Cortical collecting duct cells	Increased by aldosterone Increasing ENaC-mediated Na^+^ transportation	[39]
miR-24	*CYP11B1* *CYP11B2*	Overexpressed	Adrenal gland cells	Induced by aldosterone Reducing the production of cortisol and aldosterone	[39,40,41]
miR-34c	*CAMK2B*	Reduced	Principal cortical collecting duct cells	Reduced due to aldosterone Inducing fibrosis	[42]
miR-196b	*SOCS2*	Overexpressed	Fibroblasts	Induced by aldosterone PTEC-derived EVs taken up by fibroblasts increase ECM production	[43]
miR-766	*NR3C2*	Overexpressed	Mesangial cells	Anti-inflammatory effect via inhibition of NF-κB signaling	[44,45]
miR-466a-e	*NR3C2* *SGK1*	Overexpressed	Principal cortical collecting duct cells	Increased after 24 h of stimulation by aldosterone The negative feedback loop involved in aldosterone escape	[46]
miR-324	*NR3C2* *ELAVL1*	Overexpressed	Principal cortical collecting duct cells	Increased under conditions of hypertonicity Compromising MR signaling	[47,48,49,50]
miR-30	*NR3C2* *ELAVL1*	Overexpressed	Principal cortical collecting duct cells	Increased under conditions of hypertonicity Compromising MR signaling	[47,48,49,50]
LncRNA Tug1-miR-29a	*COL1A1* *COL3A1* *COL4A1* *COL5A1* *COL5A2* *COL5A3* *COL7A1* *COL8A1* *LTGB1* *MMP2*	Tug1: Overexpressed miR-29a: Reduced	Proximal tubular cells	Angiotensin II induces Tug1, which enhances renal fibrosis by binding to MR and suppressing miR-29b-3p.	[51,52,53,54]
miR-21	*TIMP3* *SMAD7* *PTEN*	Overexpressed	Mesangial cells (*TIMP3*) Proximal tubular cells (*SMAD7*, *PTEN*)	Increased due to DOCA-salt Increased by TGF-beta and/or HG Inducing ECM and fibrotic genes	[55,56,57,58,59,60,61,62]

ROMK, renal outer medullary potassium; 11β-HSD2, 11β-hydroxysteroid dehydrogenase type 2; ENaC, epithelial sodium channel; PTECs, proximal tubular epithelial cells; EVs, extracellular vesicles; ECM, extracellular matrix; MR, mineralocorticoid receptor; DOCA, deoxycorticosterone acetate.

## Data Availability

No new data were created or analyzed in this study. Data sharing is not applicable to this article.

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
