# Peer review of "Potential of Modulating Aldosterone Signaling and Mineralocorticoid Receptor with microRNAs to Attenuate Diabetic Kidney Disease"

_ijms, 2024, doi:10.3390/ijms25020869_

Round 1

Reviewer 1 Report

Comments and Suggestions for Authors

In the present Review entitled Potential of modulating aldosterone signaling and mineralocorticoid receptor with microRNAs to attenuate diabetic kidney disease

The Review is interesting, and I listed some comments that I think can help authors to improve their manuscript:

Major comments:

1. The authors should improve the introduction by adding more details about the pathophysiology and nature of the disease.

2. I would write: 1.1. The function of miRNA; underneath that, you may list the many miRNA types, such as 1.1.1. MiR-192.

3. The authors should clarify  the abbreviation that they used in the review and the graphical abstract.

4. The roles of miRNA need to be revised by the authors because, to me, the majority of them are not clear.

Reviewer 2 Report

Comments and Suggestions for Authors

Shinji Hagiwara et al critically examines the roles of miRNAs in modulating aldosterone signaling and mineralocorticoid receptor (MR) activation, with a focus on their implications for kidney injury, inflammation, and fibrosis. The authors propose that gaining insights into the intricate interplay between miRNAs and the RAAS pathway could pave the way for innovative strategies to target aldosterone signaling and MR activation through miRNAs. The manuscript is well-drafted, and the authors present a clear research question. The writing adheres to relevant reporting standards; however, a few suggestions are provided to enhance the manuscript:

Line 22: Consider using a more professional term than "adjusting" for miRNAs' role in aldosterone signaling.

1.1. MiR-192: Remove brackets from all subtitles for consistency.

The authors could investigate deeper into the current landscape of miRNA-based therapies in various diseases beyond the scope of the reviewed topic.

Line 403: The phrase "translation of miRNA-based interventions" is unclear; consider revising for clarity.

Consider introducing a separate section with a figure on the pathophysiology of diabetic kidney disease (DKD) to enhance organization and understanding.

The table summarizing the role of miRNA in aldosterone signaling and its relation to the mineralocorticoid receptor and renal disease may benefit from clarification. Ensure that the listed miRNAs are specific to DKD, as some may have roles in other diseases targeting different pathways. Provide justification for their relevance and specificity to DKD.

Comments on the Quality of English Language

Minor revision is required

Round 2

Reviewer 1 Report

Comments and Suggestions for Authors

We appreciate the authors work. The authors has done with all the required modifications. I accept the manuscript for publications.